# Establishment and Expression of Cytokines in a *Theileria annulata*-Infected Bovine B Cell Line

**DOI:** 10.3390/genes10050329

**Published:** 2019-04-30

**Authors:** Muhammad Rashid, Guiquan Guan, Jianxun Luo, Shuaiyang Zhao, Xiaoxing Wang, Muhammad Imran Rashid, Muhammad Adeel Hassan, Muhammad Uzair Mukhtar, Junlong Liu, Hong Yin

**Affiliations:** 1State Key Laboratory of Veterinary Etiological Biology, Key Laboratory of Veterinary Parasitology of Gansu Province, Lanzhou Veterinary Research Institute, Chinese Academy of Agricultural Sciences, Xujiaping 1, Lanzhou 730046, Gansu, China; rashidkhan.laghari@gmail.com (M.R.); guanguiquan@caas.cn (G.G.); luojianxun@caas.cn (J.L.); yicheng20081111@126.com (S.Z.); wangxiaoxing90@126.com (X.W.); uzairvetdoc@yahoo.com (M.U.M.); 2Department of Parasitology, University of Veterinary and Animal Sciences, Lahore 54200, Pakistan; imran.rashid@uvas.edu.pk; 3Department of Parasitology, Cholistan University of Veterinary and Animal Sciences, Bahawalpur 63100, Pakistan; adeelalvi21@yahoo.com; 4Jiangsu Co-Innovation Center for the Prevention and Control of Important Animal Infectious Disease and Zoonose, Yangzhou University, Yangzhou 225009, Jiangsu, China

**Keywords:** *Theileria annulata*, transformation, B cell line, cytokines, antigenic stimulation

## Abstract

This study aimed to establish a pure single-cell *Theileria annulata*-infected B cell line for the assessment of cytokine production in transformed and lipopolysaccharide (LPS)-stimulated cells. Several studies have aimed to identify cell surface markers in *T. annulata*-transformed cells; however, no information on cytokine production in these cells is available. To investigate the potential of the transformed cells to produce cytokines and their potential responses to antigen-stimulation, we purified mature B cells (CD21) from the whole blood of cattle experimentally infected with the *T. annulata* Kashi strain by magnetic separation. The purity and specificity of the established cell line was assessed by the identification of specific cell surface markers (CD21, IgM, and WC4) by flow cytometry analysis. The transcript levels of the cytokines IL1A, IL1B, IL2, IL4, IL6, IL8, IL10, IL16, LTA, TGFB1, TNFA, IFNA, and IFNB in transformed, buparvaquone (BW720c)-treated cells, and antigen-stimulated cells were analyzed by quantitative polymerase chain reaction (qPCR) using cDNA from these cells. A *T. annulata*-infected bovine B cell line was successfully established with a purity of ~98.8% (CD21). IL4 and IL12A were significantly (*p* < 0.01) upregulated in the transformed cells. In BW720c-treated transformed cells, IL12B, TGFB1, and IFNB were significantly (*p* < 0.01) upregulated. Notably, no significant (*p* > 0.05) upregulation of cytokines was observed in LPS-stimulated transformed cells. Moreover, IL1A, IL1B, IL8, and IL16 were significantly (*p* < 0.01) upregulated in LPS-stimulated B cells. Our data signify the potential use of this cell line for cytokine production, observance of immunoglobulins, and production of an attenuated vaccine against tropical theileriosis.

## 1. Introduction

*Theileria annulata*, an obligate intracellular protozoan parasite that belongs to the phylum Apicomplexa, is economically important for domestic and wild bovids [1,2]. Infection by intracellular pathogens, such as *T. annulata*, causes bovine tropical theileriosis, which is especially fatal in calves. In chronic cases, this infection can assume a subclinical form, leading to productive losses, as previously described [2]. The infection rate depends on the dose and virulence of the parasite [3]. Studies have demonstrated that follicular dendritic cells (FDCs) capture, retain, and transfer unprocessed Ag to naïve B cells for the production of an immune (antibodies) response, and signaling types of dendritic cells (DCs) can secrete immunoglobulin IgG isotypes [4,5,6]. *T. annulata* macroschizonts transform macrophages, B cells, and DCs [7,8]. *Theileria parva* was previously shown to transform T lymphocytes, but B lymphocytes were observed at a low frequency due to the pathogenicity of these parasites [8]. The host immune response to *T. annulata* is complex due to different life stages of the parasite in the host and antigenic heterogeneity. The parasite displays diverse surface antigens that necessitate a specific immune response at each life stage rather than an immune response being generated for only one stage [9]. For this reason, *T. annulata* sporozoites escape the immune response and invade and transform leukocytes while macroschizonts are able to induce host cell transformation without integrating parasite DNA into the host genome [10]. The transformation rate of B cells was previously demonstrated to be 1:6897, indicating that only one out of 6897 cells is transformable [11]. These transformed cells endlessly proliferate without the supplementation of cytokines or growth promoters [12], and this process that can be maintained indefinitely by producing the next subsequent generation of cells in fresh, complete culture medium [13]. The association of the host cell mitotic apparatus with the parasite macroschizonts enables their simultaneous division, thus ensuring the transfer of parasites to daughter cells. The exposure of infected cells to buparvaquone (BW720c) leads to parasite macroschizonts death followed by the termination of proliferation and provokes the apoptosis of host cells within a few days [14]. Immunoglobulin M (IgM), cluster differentiation 21 (CD21), and CD19-like (WC4) are typical surface markers on B cells [15,16]. However, surface marker expression was shown to be downregulated in *T. annulata*-transformed cells in several previous studies [8,10].

B cells are engaged during innate, adaptive, and humoral immune responses against exposed antigens. The host immune response consists of two main types of immune cells, T and B lymphocytes that work in concert for the production of an adaptive immune response to eliminate or destroy invading pathogens. While naïve B lymphocytes are unable to illicit an immune response, naïve T cells produce multiple cytokines (interleukins: IL7, IL4, IL6, and IL10 and interferons: IFNA, IFNB, and IFNG) upon activation. These cytokines induce the activation and differentiation of B cells, resulting in the production of their own cytokines and immunoglobulin isotypes to regulate the immune response, which plays an important role in the survival, development, and proliferation of these cells [17]. The B cell responses to both T cell-independent (TI) and T cell-dependent (TD) antigens play an integral role in the generation of protective immune responses to various pathogens.

For immunological studies, lipopolysaccharide (LPS) is widely used as an antigen stimulator of cells allow observance of their immune response in the context of cytokine and immunoglobulin production [18]. Cytokines are soluble proteins with low molecular weights (5–70 kDa). Each type of immune cell produces and releases the same or a specific type of cytokine against the antigen [18,19]. These cytokines aid in the transmittance of activation messages, chemoattraction, inhibition, and apoptosis of infected cells. Cytokines are multifunctional molecules that mediate a wide range of physiological responses and play a fundamental role in immune responses, especially in normal T cell-mediated immunity, autoimmunity, cancer, inflammatory responses, and allergies [19]. Under diverse physiologically relevant conditions, normal B cells (nBCs) exhibit distinct effector cytokine expression patterns [20]. Previous studies have reported that viral and CD40-mediated B cell stimulation results in the secretion of anti-inflammatory IL10, proinflammatory IL1B and IL4, anti- and proinflammatory IL6, tumor necrosis factors (TNFs), lymphotoxin alpha (LTA), and transforming growth factor beta1 (TGFB1) [20,21], while *T. annulata*-transformed mononuclear cells (mostly focused on macrophages) secrete IL1A, IL1B, IL2, IL6, IL10, IL12, TNFA, and IFNA [3,22]. Moreover, IL1, TNFA, IFNA, and IFNB are reportedly secreted from B cells that act on T lymphocytes and macrophages [23]. Similarly, IL8 and IL16 are secreted from lymphocytes [24]. Cytokine production in B cells infected with viruses [21] or stimulated with antigens [19,20] and in macrophages infected with *T. annulata* [3] in vitro have been investigated; however, no studies on *T. annulata*-transformed B cells (TaBCs) have been reported, and the current project was thus designed on this basis. Collectively, the IL1A, IL1B, IL2, IL4, IL6, IL8, IL10, IL12A, IL12B, IL16, TNFA, IFNA, IFNB, LTA, and TGFB1 genes were considered for this study.

The present study provides an important framework for a comprehensive overview of cytokine production in transformed cells. Additionally, this study provides insight into the identification of B cell antigen recognition receptors and the in vitro application of T lymphocyte recombinant cytokines to transformed cells to further observe their immune response.

The current study aimed to establish a *T. annulata* sporozoite-infected bovine B cell line. Surface markers for cell purity and specificity were analyzed by flow cytometry. Additionally, we tested the hypothesis that transformed and antigen-stimulated cells upregulate B cell-specific cytokine production. The expression levels of the IL1A, IL1B, IL2, IL4, IL6, IL8, IL10, IL12A, IL12B, IL16, TNFA, IFNA, IFNB, LTA, and TGFB1 cytokines were analyzed in this study using an in house developed and validated quantitative polymerase chain reaction (qPCR) assay. This tool was used to study the expression levels of the selected cytokines in established and BW720c-treated B cell lines as well as in LPS-stimulated normal and transformed cells.

## 2. Materials and Methods

### 2.1. Reagents and Antibodies

The mouse anti-bovine CD21 (MCA1424PE), mouse anti-bovine IgM (AAI19F), mouse anti-bovine WC4 (MCA1648G), and negative control mouse IgG1 (MCA928F) antibodies were purchased from Bio-Rad (Hercules, CA, USA). The PrimeScript™ RT reagent kit with gDNA Eraser (cat. no. RR047A) and SYBR^®^ Premix Ex Tq™ II (Tli RNaseH Plus) (cat. no. RR820A) were obtained from Takara, Co., Ltd. (Dalian, China). TRIzol (cat. no. 15596-026; Invitrogen, Carlsbad, CA, USA) and the pGEM T Easy vector (cat. no. A137A) were purchased from Promega (Madison, WI, USA). The anti-FITC microbeads (130-048-701), anti-PE microbeads (130-048-801), and LS column (cat. no. 130-042-401) were purchased from Miltenyi Biotec (Bergisch Gladbach, Germany). Culture medium (RPMI 1640, lot: 1930005) and fetal bovine serum (FBS, lot: 1828728) were purchased from Gibco, Life Technologies (Carlsbad, CA, USA) and LPS (cat. no. L2630) was purchased from Sigma-Aldrich (St. Louis, MO, USA).

### 2.2. Experimental Animals

*T. annulata*-free cattle (Chinese yellow cattle) <1 year of age were maintained at the animal experimental unit of the Chinese Academy of Agricultural Sciences (CAAS) at Lanzhou Veterinary Research Institute (LVRI), Lanzhou, Gansu, P.R. China in accordance with the instructions and guidelines of the animal ethics committee (permit no. LVRIAEC-2018-001), which were approved by the People’s Republic of China. For seven days, the cattle were exposed to nearly 100 adult *Hyalomma anatolicum* ticks carrying pure *T. annulata* Kashi strain pathogens that were maintained in the abovementioned laboratory. Infected cattle were kept under intensive care, and their health parameters (body temperature, lymph node swelling, confinement of ticks to the backbone by fixing a cloth bag, and thin blood smear examination) were regularly recorded on a daily basis for 13–15 days; finally, total blood was collected for merozoite isolation (used for laboratory project). After ten days of infection, blood from acutely infected cattle was collected for mature B cell (CD21) isolation, leading to the development of a transformed cell line. Furthermore, blood from pathogen-free cattle was processed for cell isolation and used as a control (∆Ct as calibrator for nBCs-LPS and TaBC), and nBCs were stimulated with LPS to induce cytokine production and compared with TaBCs. Piroplasm-free and piroplasm-infected cattle were confirmed by thin blood smear microscopic examination and PCR amplification of the target 18S rRNA gene by previously used and optimized genus-specific primers (primers details are shown in the Results section) [25].

### 2.3. Magnetic Cell Separation

Magnetic cell sorting is an effective method for the isolation of highly pure and viable cells [26]. Flow cytometry is a powerful technique for studying fluorescence-labeled surfaces as well as intracellular molecules [27]. CD21^+^ bovine mature B cells [28] were isolated by magnetic separation according to a previously described method [29,30] with minor modifications to obtain a highly pure population (>95%). Briefly, whole blood (from infected and normal cattle separately) was collected into 9 mL tubes containing K_3_EDTA by venipuncture of the jugular vein. Peripheral blood mononuclear cells (PBMCs) were isolated by density gradient centrifugation using Ficoll-Paque^TM^ plus (cat. no. 17-1440-02; GE Healthcare, Uppsala, Sweden) at a density of 1.078 g mL^−1^. The PBMCs were labeled with mouse anti-bovine antibodies (CD21) and anti-microbeads for magnetic separation via an LS column [31]. The percentage of specific cells (CD21^+^) in the PBMCs and the purity of the isolated cells were analyzed by flow cytometry. The remaining purified cells (infected and infection-free) were seeded in 24-well plates at a concentration of 2 × 10^6^ cells well^−1^ and cultured in complete medium comprising Gibco RPMI 1640 medium (L-glutamine and 25 mM HEPES), 50 µg mL^−1^ gentamicin, 50 µM 2-mercaptoethanol, 25 µg mL^−1^ amphotericin B, and 10% heat inactivated FBS at 37 °C and 5% CO_2_ for further usage. During the cell isolation process, all of the solutions used were filtered through a syringe-supported Millex^®^GP 0.22 µm filter (lot: R5SA76494; Merck Millipore Ltd., Cork, Ireland) to avoid the risk of contamination.

### 2.4. Cell Line Establishment and Maintenance

For cell line development, initially mature nBCs were isolated and co-cultured in vitro with sporozoites (grounded tick salivary glands) several times. Later, mature B cells were isolated from experimentally tropical theileriosis-infected cattle. These cells (2 × 10^6^ cells well^−1^ in 24-well plates) were maintained in complete culture medium at 37 °C and 5% CO_2_, and the medium was replaced twice a week. Cells began to proliferate at 13 days of postisolation and proliferating cells were transferred into a 25 cm^2^ culture flask for bulk production on day 23 post-isolation. These transformed cells were labeled with CD21 antibodies at the 3rd passage to identify the surface marker. To obtain a pure cell line, transformed cells were again labeled with CD21 antibodies followed by magnetic separation at the 4th generation and allowed to grow after purity analysis. A serial dilution of purified transformed cells was performed in a 96-well plate to clone a cell for homogeneous cell line development [32]. The surface markers of five clones were analyzed, and appropriate clones were selected for development of a cell line originating from a single cell and cytokine production. To further investigate the cell specificity of the established cell line, transformed cells were labeled with CD21, IgM, and WC4 surface markers antibodies at the 6th, 10th (polyclone), and 15th (single-cell clones) generations (Figure 1), and then analyzed by flow cytometry.

### 2.5. Cytotoxicity Assay

The MTT assay is the colorimetric assay for assessing the cell metabolic activity. This assay was used for optimization of LPS and BW720c concentrations to evaluate the immune response in antigen-stimulated and treated cells. For this assay, 10 µL (5 µg mL^−1^ PBS) of MTT solution was added to each well during the final 6 h of incubation. After this incubation period, the plates were centrifuged at 400 *g* for 8–10 min, and the supernatant was collected. DMSO (100 µL) was added to each well, and the plates were then shaken to ensure that all crystals were dissolved. The amount of MTT formazan produced during incubation was measured by an ELISA reader at a reference wavelength of 630 nm and a test wavelength of 550 nm (OD_550_) [19]. The simulation index (SI) was calculated as follows:SI = (mean OD of stimulated/treated cells − mean OD of blank)/mean OD of un-stimulated cells

### 2.6. Antigenic Stimulation

The stimulation was performed to compare the efficacies of cytokine production in normal and transformed cells and the abilities of these cells to recognize antigens. The nBCs and TaBCs were plated in six-well culture plates at 5 × 10^6^ cells well^−1^ and stimulated with 1 µg mL^−1^ LPS in 3 mL culture medium for 24 h at 37 °C and 5% CO_2_ to induce significant cytokine production, while control cells remained untreated [33]; the cells were then washed with PBS for mRNA extraction by the TRIzol method as described below in detail.

### 2.7. Theileriacidal Treatment

BW720c, a hydroxynaphthoquinone that likely acts as a ubiquinone analog [34], specifically blocks electron transport in parasites. The transformed cells were treated with the theileriacidal drug to evaluate the regulation of cytokine production in the absence of schizonts. TaBCs were incubated with 100 ng mL^−1^ BW720c for 48 h at 37 °C and 5% CO_2_ [33,35] and then washed prior to mRNA extraction according to the method described below. The mRNA was processed to quantify the quantification of the transcription levels of the various cytokines analyzed in this study.

### 2.8. Flow Cytometry Analysis

For cell purity and surface marker identification, antibody-labeled cells were washed three times with ice-cold PBS containing 0.2% Tween 20. The fluorescence intensities of surface markers (CD21^+^, IgM^+^, and WC4^+^) were analyzed on the BD Accuri C6 instrument (Becton Dickinson and Company 1, Becton Drive Franklin Lakes, NJ, USA), while untreated PBMCs and TaBCs were used as background fluorescence controls [36]. Labeled cells were gated by side and forward scatter characteristics. For each sample, 10,000 events were collected on the FACSCalibur instrument (Accuri C6) for flow cytometry analysis. Gating of the live target cells was performed by selecting the main cell populations in the forward and side scatter profiles. Normalized mean florescence was computed by subtracting of the geometric mean florescence of the cells that were not incubated with any florescence labels [37].

### 2.9. PCR and Sequencing Analysis

Bovine cytokines, consisting of IL1A, IL1B, IL2, IL4, IL6, IL8, IL10, IL12A, IL12B, IL16, TNFA, IFNA, IFNB, LTA, and TGFB1, reported by [17,20,21,23,24,38], were considered in this study. Reference sequences used for primer design were obtained from a public database (National Center for Biotechnology Information, NCBI). Primers used for the amplification of short cytokine gene segments were designed using the GenScript online browser [39] and NCBI [40], and melting temperatures (less than 2 °C difference) and self-dimers (less than 10 kcal/mole) were evaluated with an online browser [41]. The primers that met the criteria were selected and synthesized by Sangon Biotech Co., Ltd. (Shanghai, China). The target genes were amplified, purified, and cloned into the pGEM ^TM^-T Easy vector. Briefly, PCR products were separated on 4% agarose gels and the desired bands were purified using a gel/PCR extraction kit (Biomiga, Inc., San Diego, CA, USA). The purified PCR products were then ligated into the pGEM^TM^-T Easy vector and transformed into competent cells (DH5α) [42]. The bacterial culture samples were sent to Sangon Biotech Co., Ltd. for sequencing.

### 2.10. RNA Extraction and cDNA Synthesis

RNA was extracted from nBCs, TaBCs, BW720c-TaBCs and LPS-stimulated nBCs and TaBCs using TRIzol reagent. Briefly, each cell type was harvested in a separate 1.5 mL tube by centrifugation at 12,000 *g* for 2 min and then washed with 1 × PBS. For total RNA isolation, the TRIzol method was used as previously described [43]. The concentration and purity (260/280 and 26/230 nm ratios) of isolated RNA was determined on a NanoDrop spectrophotometer (Thermo Scientific 2000/2001, Wilmington, DE, USA). cDNA was synthesized from 1 µg of RNA using the PrimeScript™ RT reagent kit with gDNA Eraser according to the manufacturer’s instructions and stored at −20 °C until further experimental usage.

### 2.11. Quantitative PCR Analysis

qPCR is an extremely sensitive technique that allows the quantification of rare transcripts and the identification of slight changes in gene expression from a limited sample quantity [44]. qPCR was performed using SYBR^®^ Premix Ex Tq™ II (Tli RNaseH Plus) according to the manufacturer’s instructions. Briefly, the 20 µL reaction mixture comprised 10 µL of SYBR^®^ Premix Ex Taq II, 0.8 µL (16 µM) of each primer (forward and reverse), 0.4 µL of Rox reference dye II (50×), 2 µL of cDNA, and 6 µL of RNase-free water. The reactions were performed on the Mx3005P qPCR Systems™ (Agilent Mx3005P; Agilent Technologies, Santa Clara, CA, USA) in three repeating steps/segments as follow: denaturation at 95 °C for 30 s; 40 cycles of annealing at 95 °C for 5 s and 60 °C for 34 s, followed by extension at 95 °C for 15 s, held at 60 °C for 1 min and held at 95 °C for 0.15 s according to the SYBR^®^ Premix Ex Tq™ II kit instruction. cDNA was serially diluted (1/10) prior to qPCR analysis to optimize the cytokine primers. The qPCR data were analyzed by regression (known cDNA quantities) based on the threshold cycle (Ct) for each cytokine to calculate *R*^2^ values [45]. All reactions were run in triplicate, and three samples of each gene were run in each qPCR assay to analyze the gene expression levels in transformed, treated and stimulated cells. The Ct values of β-actin were used for the calculation of ∆Ct, and the ∆Ct of BCs (normal and transformed) were used for the calculation of ∆∆Ct, termed the normalizer and calibrator, respectively. The fold change values were calculated by the 2^−∆∆Ct^ formula as previously described [46].

### 2.12. Data Analysis

Sequence data were analyzed by EditSeq, SeqMan, and MegAlign (DNA Star, Madison, WI, USA). Flow cytometry florescence data were analyzed by FlowJo X version, 10.0.7 (BD Biosciences, Franklin Lakes, NJ, USA). Fold change values were calculated by the 2^^−∆∆Ct^ method using Microsoft Excel 2010 (Redmond, Washington, USA). One-way ANOVA, multiple comparisons (fold change in nBCs-LPS for TaBCs and TaBCs in TaBCs-BW720c and TaBCs-LPS as the reference to analyze significant values), and graphical analyses were performed using GraphPad Prism software version 7 (La Jolla, CA, USA). Significant values obtained are presented as * *p* < 0.05, ** *p* < 0.01, and *** *p* < 0.001; ns represents non-significance (*p* > 0.05).

## 3. Results

### 3.1. Confirmation of Experimental Animals for Piroplasmosis

Infected and infection-free experimental animals as well as the established cell line were confirmed using thin blood smear microscopic examination (Figure 2) and a genus-specific pair of primers, (forward) 5′-AAGCCATGCATGTCTAAGTAGAAGCTTTT-3′ and (reverse) 5′-GAATAATTCACCGGATCACTCG-3′ which produced a 1586 bp product at an annealing temperature of 57 °C for 1.5 min and a final extension at 72 °C for 10 min (Figure 3). The sequence obtained was submitted to the NCBI database under accession number: MK415058. Cells isolated from infection-free cattle were used as controls, while cells isolated from infected cattle were processed for cell line development.

### 3.2. Established B-Cell Line

The purity of magnetically isolated B cells (CD21^+^) and the percentage of B cells in PBMCs were ~99 and ~32%, respectively (Figure 4). The cells purified from piroplasm-free cattle were unable to transform after in-vitro infection with sporozoites. Hence, B cells were isolated from experimentally infected cattle for culture until transformation and for cell line maintenance (culture conditions are described in the materials and methods section under the same subheading). Transformation of the infected cells began on the 13th day postisolation and incubation. CD21 surface markers were identified in 83% of these transformed cells at 3rd passage, and the remaining 17% were considered efficiently transformed but impure mononuclear cells (<1%) (Figure 5). To obtain a highly pure cell population, 4th generation transformed cells were labeled with CD21 antibodies for magnetic cell separation, and a purity of ~99% was achieved. From 5 clones, one expressing CD21 (98.8%), IgM (3.51%), and WC4 (0.51%) was selected for establishment of cell line from a single cell and cytokine production. These newly established B cell lines were maintained in complete culture medium at 37 °C and 5% CO_2_ and passaged twice a week according to their proliferation growth rate. These cell lines were preserved at the 8th (polyclonal) and 18th (single-cell clone) generations and stored at the Vector and Vector Borne Disease Laboratory, LVRI, Lanzhou, for further experimental purposes.

### 3.3. Specificity Analysis of the B Cells Line

The percentages of the CD21, IgM and WC4 surface markers in 6th and 10th generation cells (polyclonal cell line) were ~99.0, 6.52–14.2 and 1.3–2.5%, respectively, whereas those of IgM and WC4 in normal isolated pure B cells were 96.6 and 85.9%, respectively. The percentages of CD21 (98.8, 99.7, 98, 98.5, and 97%; mean ± SD of 98.4 ± 1.00), IgM (3.51, 0.64, 1.42, 0.99, and 2.83%; mean ± SD of 1.88 ± 1.23) and WC4 (0.51, 1.96, 0.71, 0.80, and 0.97%; mean ± SD of 0.99 ± 0.57) were determined in the five clones, respectively. Of these, clone 1 was maintained, preserved, and used for the identification of cytokine production (Figure 6).

### 3.4. Optimized Conditions for BW720c and LPS

Cells in a wide number of ranges were cultured with various concentrations of LPS and BW720c for 24 and 48 h, respectively. As determined by MTT assay, the optimum concentrations of LPS and BW70c for the maximum viability of 5 × 10^6^ cells well^−1^ were 1 µg mL^−1^ and 100 ng mL^−1^, respectively, and these concentrations were used in later experiments to examine cytokine production in BW720c-treated and LPS-stimulated cells.

### 3.5. Cytokine Primer Optimization

Target genes were amplified from DNA and cDNA extracted from TaBC with specific primers (Figure 7), purified, ligated into the pGEM T Easy vector, and transformed into DH5α competent cells. These genes were finally confirmed by sequencing analysis for each cytokine evaluated in this study. The optimized primers with correctly sequenced upon amplification and their *R*^2^ values (Table 1; Table 2; Figure 8) were used for qPCR to evaluate the transcript levels of the abovementioned genes in transformed and theileriacidal-treated TaBCs as well as in LPS-stimulated nBCs and TaBCs. Additionally, the primer sequences for IL8, IL16, LTA and TGFB1 were acquired from previously published literature [21,45,47].

### 3.6. Interleukins

To quantify the cytokine levels, samples from TaBCs, BW720c-treated transformed B cells, and LPS-stimulated nBCs and TaBCs were acquired at the 16–18th generations for cDNA synthesis (Figure 9).

In this study, IL1A, IL1B, IL2, IL4, IL6, IL8, IL10, IL12A, IL12B, IL16, LTA, and TGFB1 [38] were considered to be the cytokines likely produced and to interact with bovine B cells (Figure 10). There was no significant difference in cytokine production between the mono—and polyclonal B cell lines of same passage, indicating that these derived from the same origin, which was confirmed by surface marker expression in these cell lines. Thus, only cytokines of monoclonal B cell line are mentioned here while polyclonal are provided in Appendix A. In TaBCs (∆Ct of nBCs was used as the calibrator for the ∆∆Ct calculation), IL1A, IL1B, IL4, IL6, IL10, and IL12A were upregulated (fold change >1.0) on average by 6.33 ± 1.55-, 1.09 ± 0.51-, 10.57 ± 2.67-, 9.61 ± 7.91-, 3.44 ± 1.35-, and 18.62 ± 12.55-fold, respectively. In BW720c-treated transformed cells (∆Ct of TaBCs was used as the calibrator for the ∆∆Ct calculation), the cytokines IL6, IL8, IL10, IL12A, IL12B, and TGFB1 were upregulated on average by 4.96 ± 1.23-, 1.77 ± 0.22-, 1.92 ± 0.71-, 20.60 ± 0.72-, 4.45 ± 2.50-, and 0.54 ± 0.26-fold respectively, while the others were downregulated. In addition, the expression of all cytokines was downregulated in TaBCs stimulated with LPS (∆Ct of TaBCs was used as the calibrator for the ∆∆Ct calculation) except for IL6, which was upregulated by 1.15 ± 0.21-fold on average. However, in LPS-stimulated nBCs, cytokines IL1A, IL1B, IL6, IL8, IL10, IL12A, and IL16 were upregulated (∆Ct of nBCs was used as the calibrator for the ∆∆Ct calculation) on average by 44.41 ± 12.87-, 389.36 ± 83.73-, 11.38 ± 5.38-, 4.36 ± 1.73-, 2.30 ± 1.16-, and 2.08 ± 0.92-fold, respectively (Figure 11, Table 2).

### 3.7. Interferons

In transformed and LPS-stimulated B cells, the transcript level of IFNA and IFNB were not upregulated. Moreover, only IFNB was upregulated by 3.67 ± 1.66-fold on average in BW720c-treated transformed cells. Similarly, IFNA and IFNB were upregulated by 1.08 ± 0.72- and 2.12 ± 0.98-fold, respectively, in LPS-stimulated transformed cells (Figure 11, Table 2).

### 3.8. Tumor Necrosis Factor

TNFA is a cellular signaling protein that is involved in acute-phase reactions and systematic inflammation [49], and fever is a clinical symptom in infected organisms [50]. Schizont-infected cells also secrete TNFA for the stimulation of other leukocytes.

TNFA was upregulated by only 2.91 ± 1.85- and 1.55 ± 0.96-fold in LPS-stimulated nBCs and TaBCs, respectively (Figure 11, Table 2).

## 4. Discussion

Neonatal bovine B cells express CD5 [51], whereas mature B cells express the CD21 surface marker [28]. Previously, established B cell lines were developed from in vitro infection of these cells (CD5 and CD21) with *T. annulata* sporozoites for the identification of various cell surface markers [8,35]. However, in the current study, a single transformed cell was cloned to establish a cell line and quantify significant cytokine production. During this study, mature B cells (CD21) were unable to infect sporozoites after in vitro infection due to three possible reasons. First, lymphocytes are not directly exposed to antigens and antigens are instead presented by antigen-presenting cells (APCs), such as DCs and macrophages [6]. Second, elastin (hexapeptide) receptors, which are expressed in mainly immature monocytes, but not in B and T lymphocytes, are the potential ligands for sporozoite entry, which explains the previously observed low transformation rate (1:6897) of B cells [11]. Third, some animal breeds, such as those used as experimental animals in this study, may be less susceptible to tropical theileriosis [52]. Hence, B cells (CD21^+^) from in vivo *T. annulata*-infected cattle were isolated and cultured to develop a pure cell line. The purity and specificity of the established cell line was confirmed after the identification of three different but specific B cell surface makers on TaBCs and nBCs, ensuring that the established cell line originated purely from bovine B cells. However, there are no published data on the purity of this type of cell line after confirmation of its establishment other than the scarce information about the ability of *T. annulata* to transform B lymphocytes. Except for CD21, these surface markers were consistently downregulated in later passages/generations of the cell line. These findings are supported by previously reported studies in which the expression of transformed cell surface markers rapidly decreased [7,53], but coexpression of the CD21 marker was observed following infection [35].

The host immune response to any pathogen results from a complex network of various immune effector cells and their signals [17]. Cytokines constitute a considerable portion of this signaling cascade for stimulation, and the balance of various cytokines has an intense effect on the outcomes of infectious challenges. Interestingly, *T. annulata* hijacks the host immune response to invade and transform leukocytes [54]. Previous studies claimed that transformed cells (macroschizonts containing macrophages) secrete various types of cytokines [3], which is why a set of primers for each cytokine (mentioned above) was optimized for qPCR analysis to evaluate the cytokine profiles in the TaBCs, BW720c-treated cells and LPS-stimulated cells in this study. In transformed cells, the transcript levels of IL2, IL4, IL10, and IL12A were upregulated compared with those in LPS-stimulated nBCs, but only IL4 and IL12A were statistically significant (*p* < 0.01 and *p* = 0.01), which supports our hypothesis. Cytokines such as IL4 directly stimulate B cells for the production of IgG1 and IgE [55]. Moreover, IL4 also functions as a survival factor for B and T lymphocytes, and IL12 is responsible for the induction of Th1 cell differentiation. Likewise, significantly upregulated IL4 might lead to the secretion of immunoglobulin from transformed cells. These results are in consistent with previously reported findings that transformed *T. annulata* cells secrete IL1A, IL1B, IL6, IL10, IL12, TNFA, and IFNA [3,56]. Epstein Barr virus (EBV)-transformed human B cells constitutively secrete IL1, leading to T lymphocyte proliferation [57]. In this study, IL2 expression was downregulated in all cell types due to its origination from cells other than B cells. However, other studies have claimed that transformed cells secrete IL2 to serve as a growth factor and enhance cell proliferation [22]. Exposure of infected cells to BW720c induced to parasite death followed by cessation of host cell proliferation and apoptosis within a few days [14]. Furthermore, the transcript levels of IL8, IL12B, TGFB1 IFNA, and IFNB were upregulated in BW720c-treated TaBCs compared with those in untreated transformed cells, but only IL12B and IFNB were statistically significant. LTA and TGFB1 were significantly (*p* < 0.01) produced in nBCs and TaBC-BW720c treated cells, respectively. TNF, LTA, and TGFB1 reportedly play important roles in the differentiation and proliferation of B cells, and these cytokines may exert an autocrine effect on B cells or modulate their interaction with other cell types, such as helper T cells or FDCs [21]. Similarly, the expression levels of IL1A, IL1B, IL6, IL8, IL16, LTA, IFNA, and TNFA were upregulated in LPS-stimulated nBCs compared with those in TaBCs, and the upregulation of IL1A, IL1B, IL8, and IL16 was significant. These findings are in agreement with the reported findings that effector B cells significantly produce cytokines [51]. Interestingly, no statistically significant upregulation was found in LPS-stimulated TaBCs, which was opposite from our proposed hypothesis. Fundamentally, immune responses substantially relate to the recognition of exogenous antigens by Toll-like receptors (TLRs), B cell receptors (BCRs), and CD40, which detect different pathogen-associated molecular patterns [38]. After antigen recognition, immune cells generate a wide range of cellular responses, including the secretion of anti—and proinflammatory cytokines, chemokines, and type I interferons [18]. Evidence from the literature and our own findings indicate that transformed cells regress most of the surface receptors [7], which might include TLRs, BCRs, and CD40. This downregulation potentially accounts for why antigen (LPS) was not recognized and no immune response (cytokine production) was generated by these cells.

Upon activation, naïve B cells do not secrete any cytokines, while naïve T cells initiate cytokine production upon activation, which is the inherent difference between these cell types; B cells require additional activation signals to become cytokine producers. These signaling pathways are provided by immune cells in the surrounding and B cell-specific differentiation stages. Once B cells acquire the ability to produce cytokines, they become capable of cross-regulating responses through inhibition/polarization that can negatively regulate the entire host immune response system. LPS- and cytokine (IL5 and IL10)-stimulated B cells [38] are associated with the production of natural immunoglobulins (IgM and IgG) in response to antigens [58,59,60]. These associations served as the basis for using LPS to stimulate nBCs and TaBCs to compare their cytokines production abilities and further investigate immunological changes (immunoglobulin production) in these cell types. Other immune cell cytokines are key factors for switching the immunoglobulin and their isotype production from B lymphocytes [45,55]. Hence, this study provides a basis for future studies to observe the effects of various supplemental recombinant cytokines from other immune cells on these transformed cells. The effects of the direct exposure of other immune cells and their cytokines on these transformed cells to trigger their own significant production or the switching of immunoglobulins and their isotypes were not studied herein.

## 5. Conclusions

We successfully established *Theileria annulata*-infected poly- and monoclonal B cell lines. Significant cytokine production was demonstrated in this cell line, which can now be used in vaccines/therapy against pathogens. This established pure cell line can also be used in various types of in vitro immunological studies.

## Figures and Tables

**Figure 1 genes-10-00329-f001:**
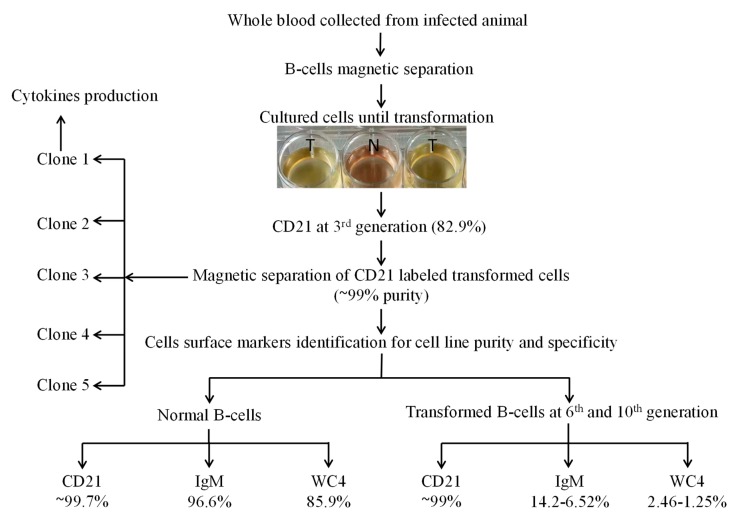
Schematic diagram for establishment of the *Theileria annulata*-transformed bovine B cell line. Here, N represents normal B cell culture, and T represents the infected cell culture wells used for transformation and proliferation for establishment of cell line.

**Figure 2 genes-10-00329-f002:**
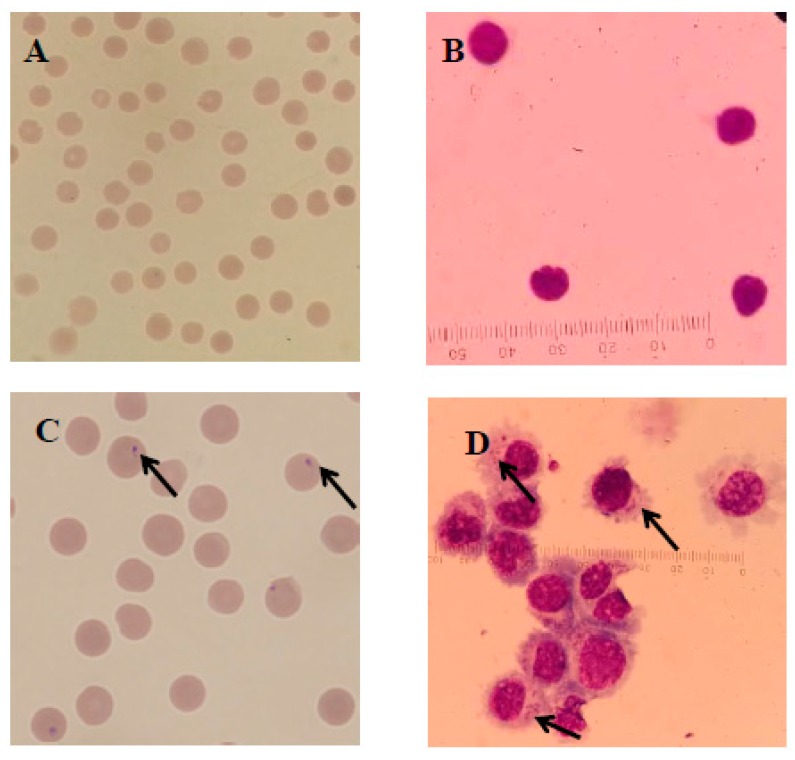
Microscopic examination of *T. annulata* merozoites in red blood cells (RBCs) and macroschizonts in transformed B cells as well as normal cells stained with Giemsa observed under a compound microscope at 1000×. Here, (**A**,**B**) represent the normal RBCs and lymphocytes, respectively, while the arrows in (**C**,**D**) show the presence of merozoites in the RBCs of infected cattle and macroschizonts in the established cell line, respectively.

**Figure 3 genes-10-00329-f003:**
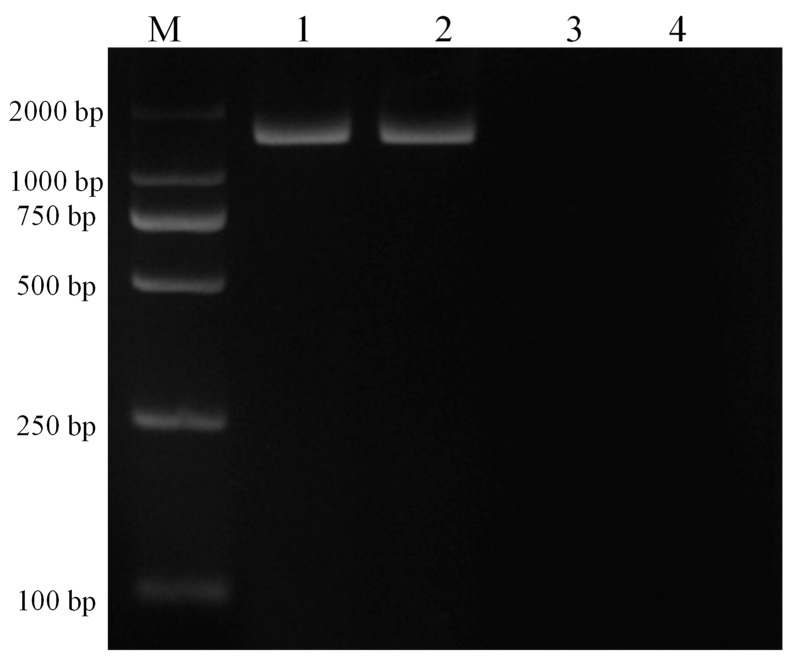
PCR amplification of 18S RNA with genus-specific primers for confirmation of piroplasm infection in experimental animals and the established cell line. Here, M represents the DNA marker (DL2000), lane 1 depicts the experimentally infected cattle with sporozoites, lane 2 depicts the transformed cells, lane 3 depicts the piroplasm free cattle, and lane 4 depicts the negative control.

**Figure 4 genes-10-00329-f004:**
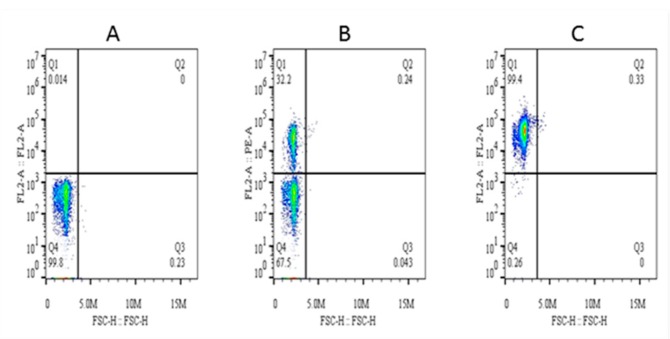
Percentage of CD21^+^ cells in peripheral blood mononuclear cells (PBMCs) and analysis of their purity after magnetic separation. Here, (**A**) is the background control, (**B**) is the percentage of CD21^+^ cells in PBMCs, and (**C**) is the purity of magnetically isolated cells.

**Figure 5 genes-10-00329-f005:**
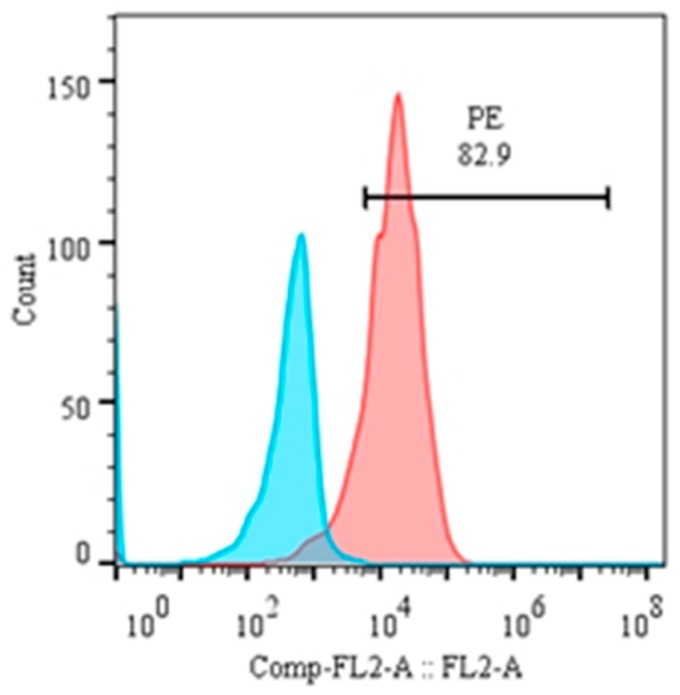
Percentage of CD21 in transformed B cells at the 3rd passage.

**Figure 6 genes-10-00329-f006:**
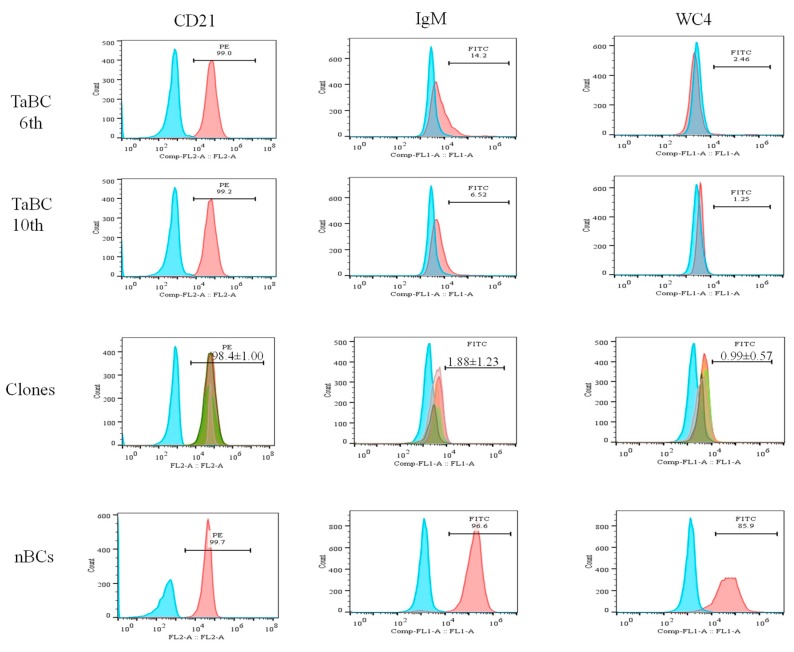
Percentages of B cell-specific surface markers (CD21, IgM, and WC4) present in transformed cells (6th and 10th generations), single-cell line clones (approximately 15th generation), and normal purified cells. The percentages of CD21 (98.8, 99.7, 98.0, 98.5, and 97%; mean ± SD of 98.4 ± 1.00), IgM (3.51, 0.64, 1.42, 0.99, and 2.83%; mean ± SD of 1.88 ± 1.23) and WC4 (0.51, 1.96, 0.71, 0.80, and 0.97%; mean ± SD of 0.99 ± 0.57) in five clones were determined. CD21 was present in mono- and polyclonal cell lines, while negligible percentage of IgM and WC4 were present.

**Figure 7 genes-10-00329-f007:**
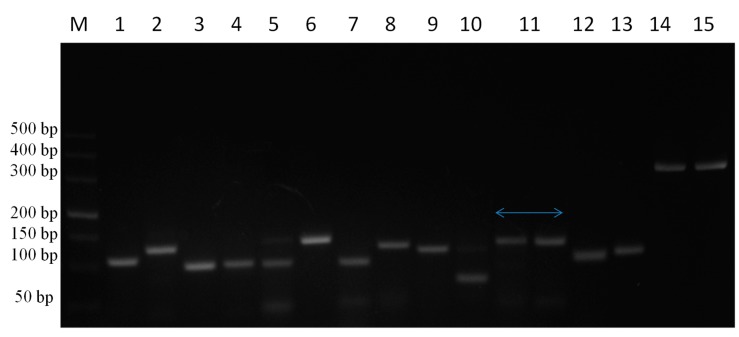
PCR amplification of cytokine genes from TaBCs DNA and cDNA using specific primer pairs. Here, M represents the DNA marker (DL500), while lines 1–15 indicate IL1A, IL1B, IL2, IL4, IL6, IL8, IL10, IL12A, IL12B, IL16, TNFA, IFNB, IFNA, LTA, and TGFB1, respectively, amplified from DNA and cDNA harvested from of transformed B cells.

**Figure 8 genes-10-00329-f008:**
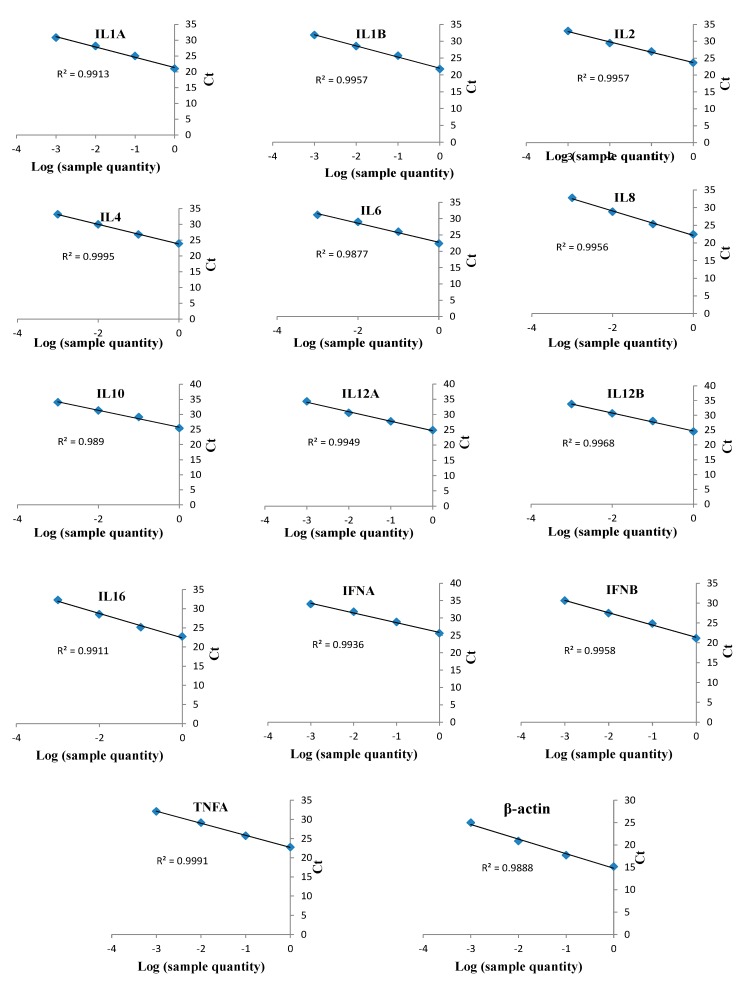
qPCR linearity measurements of bovine cytokines in cDNA from *T. annulata*-transformed B cells (TaBCs) and LPS-stimulated peripheral blood mononuclear cells (PBMCs). qPCR was performed on serially diluted (1/10) cDNA to detect and optimize the efficacies of various cytokine primers. These optimized primers were used to evaluate the transcript levels of cytokine in TaBCs, BW720c-treated transformed B cells, and LPS-stimulated normal B cells (nBCs) and TaBCs. The qPCR data were analyzed by regression analysis based on log values of sample quantity and threshold cycles (Ct) for each cytokine. The monitored cytokines and *R*^2^ values of each regression line are shown.

**Figure 9 genes-10-00329-f009:**
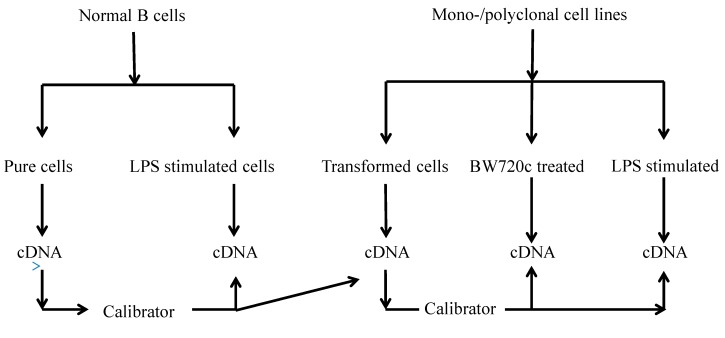
Flow sheet diagram of samples collected from various cells types and the calibrators for cytokine production analysis.

**Figure 10 genes-10-00329-f010:**
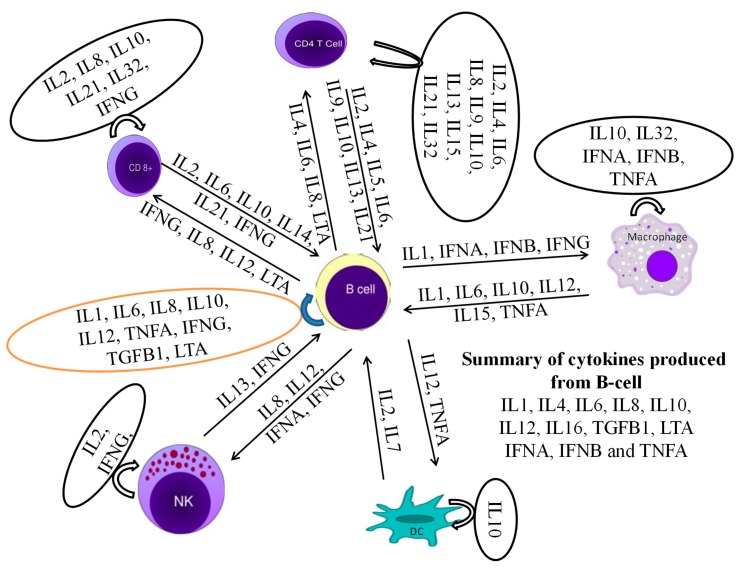
A cytokine production network as well as interaction with themselves and other leukocytes is required for the stimulation of B cells to produce their own cytokines.

**Figure 11 genes-10-00329-f011:**
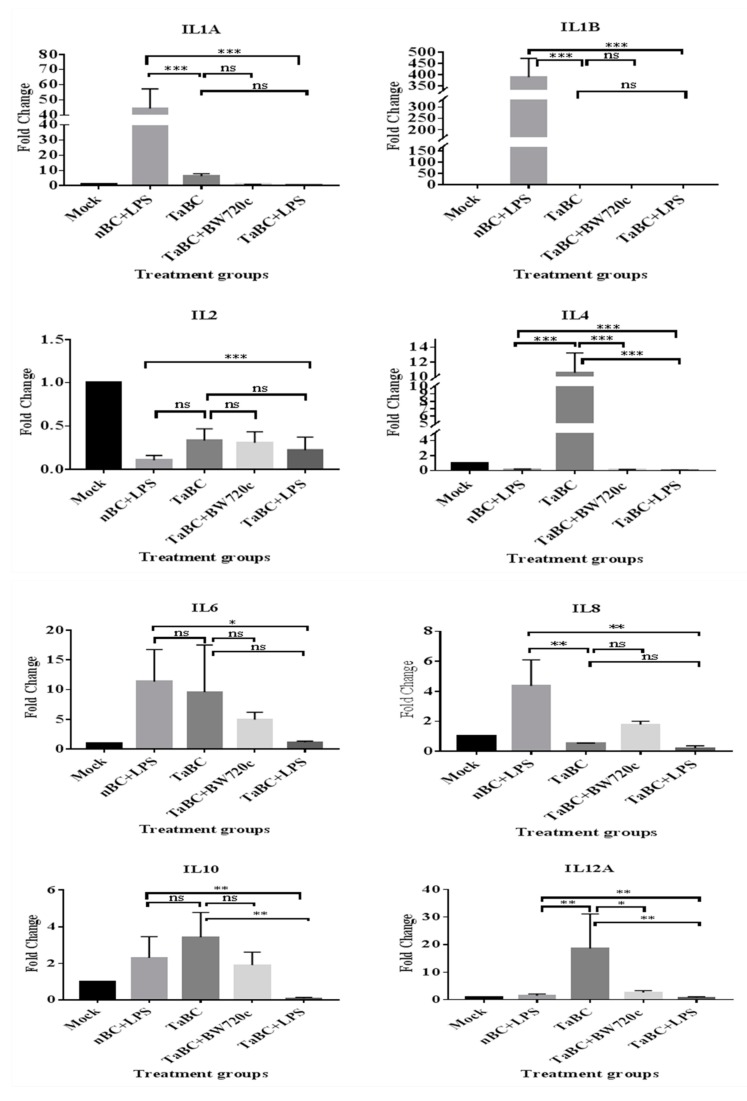
Transcript levels of cytokines in monoclonal *T. anuulata*-transformed B cells (TaBCs), BW720c-treated transformed cells, and LPS-stimulated TaBCs and normal B cells (nBCs). Here, *** *p* < 0.001, ** *p* < 0.01, * *p* < 0.05, and ns is not significant.

**Table 1 genes-10-00329-t001:** Details regarding the qPCR primers used for the quantification of cytokine production in this study.

NCBI Accession No.	Gene Name	Sense (5′ to 3′)	Primers	Product Length (bp)	Reference
NM_174092.1	IL1A	F	AGCTATGAGCCACTTCGTGA	110	This study
		R	GCCACCATCACCACATTCTC		
KX013245.1	IL1B	F	CCTTCCCTGCATTAGTGCTT	129	
		R	AGGCTGGCTTTGAGTGAGTA		
NM_180997.2	IL2	F	CCTCAACTCCTGCCACAATG	100	
		R	CCCGTAGAGCTTGAAGTAGGT		
NM_173921.2	IL4	F	GTGCTGGTCTGCTTACTGGT	102	
		R	CGTTGTGAGGATGTTCAGCG		
NM_173923.2	IL6	F	ACGAGTGGGTAAAGAACGCA	101	
		R	GAGCCCCAGCTACTTCATCC		
XM_027544730.1	IL8	F	TGGGCCACACTGTGAAAAT	136	[47]
		R	TCATGGATCTTGCTTCTCAGC		
NM_174088.1	IL10	F	TCAGCACTACTCTGTTGCCTG	100	This study
		R	GGCTGGTTGGCAAGTGGATA		
NM_174355.2	IL12A	F	ACAACCCTGTGCCTTAGAAGT	126	
		R	TGCCAGCATGTTCTGGTCTA		
NM_174356.1	IL12B	F	ACCAGAGCAGTGAGGTCTTG	116	
		R	GAGTGAACGACTCAGAGCCT		
AF143203.1	IL16	F	GAGGGCGGTCCCAGAAGT	73	[45]
		R	CTCTCTAGATGCAGTCTGTCGTTTGT		
NM_001013401.2	LTA	F	TGACACCACCTGGACGTCTC	294	[21]
		R	GGAGGGAATTGTTGCTCAGA		
M36271.1	TGFB1	F	AGAGAGGAAATAGAGGGCTT	306	
		R	ATGAATCCACTTCCAGCCCA		
Z46508.1	IFNA	F	GTGGCAGCCAGTTACAGAAG	127	This study
		R	CATAGCTTGTCCAGGAGGCT		
EU276065.1	IFNB	F	TGCCTGAGGAGATGAAGCAA	100	
		R	TCTCTGGTGAGAATGCCGAA		
GU129693.1	TNFA	F	GGCCAGGATGTGGAGAGTAG	132	
		R	CCATGAGGGCATTGGCATAC		
XM_019987862.1	B-actin	F	GGCATCCTGACCCTCAAGTA	102	[48]
		R	CACACGGAGCTCGTTGTAGA		
KF559356.1	Ta18s	F	AAGCCATGCATGTCTAAGTAGAAGCTTTT	1586	[25]
		R	GAATAATTCACCGGATCACTCG		

**Table 2 genes-10-00329-t002:** Linearity measurements from regression analysis, mean fold changes, and standard deviation (SD) values of cytokines produced in *T. annulata*-transformed (TaBCs), BW720c-treated transformed cells, and LPS-stimulated normal B cells (nBCs) and TaBCs.

Cytokines	Slope (-)	*R* ^2^	Cultured Cell Groups			
			nBC-LPS Mean ± SD	TaBCs Mean ± SD	TaBCs-BW720c Mean ± SD	TaBC-LPS Mean ± SD
IL1A	3.247	0.9913	44.41 ± 12.87	6.33 ± 1.55	0.80 ± 0.10	0.30 ± 0.05
IL1B	3.308	0.9957	389.36 ± 83.73	1.09 ± 0.51	0.19 ± 0.11	0.57 ± 0.19
IL2	3.062	0.9957	0.11 ± 0.06	0.33 ± 0.14	0.31 ± 0.13	0.22 ± 0.15
IL4	3.11	0.9995	0.13 ± 0.08	10.57 ± 2.67	0.08 ± 0.05	0.05 ± 0.03
IL6	2.934	0.9877	11.38 ± 5.38	9.61 ± 7.91	4.96 ± 1.23	1.15 ± 0.21
IL8	3.46	0.9956	4.36 ± 1.73	0.52 ± 0.03	1.77 ± 0.22	0.21 ± 0.15
IL10	2.95	0.9890	2.30 ± 1.16	3.44 ± 1.35	1.92 ± 0.71	0.09 ± 0.06
IL12α	3.112	0.9949	1.51 ± 0.58	18.62 ± 12.55	2.60 ± 0.72	0.76 ± 0.33
IL12β	3.05	0.9968	0.59 ± 0.34	0.39 ± 0.53	4.45 ± 2.50	0.75 ± 0.59
IL16	3.188	0.9911	2.08 ± 0.92	0.12 ± 0.05	0.32 ± 0.08	0.66 ± 0.33
LTA	ND	ND	0.32 ± 0.08	0.27 ± 0.05	0.22 ± 0.05	0.05 ± 0.02
TGFB1	ND	ND	0.08 ± 0.07	0.31 ± 0.06	0.54 ± 0.26	0.03 ± 0.03
TNFA	3.154	0.9991	2.91 ± 1.85	1.55 ± 0.96	0.78 ± 0.43	0.52 ± 0.17±
IFNA	2.812	0.9936	0.66 ± 0.58	0.44 ± 0.26	0.59 ± 0.23	1.08 ± 0.72
IFNB	3.105	0.9958	0.20 ± 0.11	0.28 ± 0.16	3.67 ± 1.66	2.12 ± 0.98

Here, ND (not done) depicts values that were already optimized in the literature and used in this study without optimization.

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
