# Peer review of "Establishment and Expression of Cytokines in a Theileria annulata-Infected Bovine B Cell Line"

_genes, 2019, doi:10.3390/genes10050329_

Round 1
Reviewer 1 Report
There are several aspects of editing and article construction that I want to mention. Firstly, your information, although valuable, is presented unclear, with many details which make difficult to understand the essence of the subject. Moreover, your article need a revision of English style. For example, in English language is, sometimes, an excess of comma using, but we have to respect it. For example, when you have two variables, x and y, you do not put comma between them, but when you have three and more than three variables, between penultimate and the last you have to use comma (x, y, and z).
Author Response
Revision report of reviewer 1:
Comments and Responses:
The entire questions are welcomed and answered accordingly. All major and minor changes are mentioned in red color text in the main text file. We will be available for any further clarification.
Point 1:
There are several aspects of editing and article construction that I want to mention. Firstly, your information, although valuable, is presented unclear, with many details which make difficult to understand the essence of the subject.
Response: The MS is extensively revised to clarify the meanings of each sentence and section.
Point 2:
Moreover, your article need a revision of English style. For example, in English language is, sometimes, an excess of comma using, but we have to respect it. For example, when you have two variables, x and y, you do not put comma between them, but when you have three and more than three variables, between penultimate and the last you have to use comma (x, y, and z).
Response: Before submission the article was linguistically improved by American journal experts according to advisor recommendation and institute regulation but unfortunately the improvement was not significant. We claimed to re-edit the whole text file that was well improved and an editing certificate is included within revision submission files. We hope that you can find it better now.

Reviewer 2 Report
In this study, the authors generated a Theileria annulata-schizont infected, cloned B cell line, and used that line to determine the cytokine production profile of Theileria annulata infected B cells.
My main reservation about this study is that the results were generated using a cloned line. In the closely related organism, T. parva, studies have shown that different T. parva infected, cloned lines produce different sets of cytokines. Thus, it is very likely that the results obtained by the authors using one B cell clone do not apply to T. annulata-infected B cells in general, especially in vivo, where infected cells interact with the rest of the immune system. Results would be strengthened if a large number of clones were studied, or if a several polyclonal B cell lines were analyzed.
The rationale for the experiments using LPS is unclear, and the discussion of the results for these sections does little to put the information into perspective. Extensive revision is needed to clarify why these experiments were done, and what the results contribute to the scientific community in terms of immunology, parasitology, vaccinology, etc.
The grammar needs moderately extensive revision.
Other specific comments are below:
The abstract in the manuscript file does not match the one provided on the reviewer form.
Line 51 is incorrect. Theileria parva infects both B and T lymphocytes.
Line 90 is incorrect. IL-10 is generally regarded as an anti-inflammatory cytokine, and IL-6 behaves in both pro-inflammatory and anti-inflammatory ways.
Line 100: How can evaluation of a single, clonal, line provide a comprehensive overview of cytokine production by T. annulata infected B-cells?
Section 2.2: Please include information on veterinary care provided to infected animals and severity of clinical signs exhibited.
Section 2.4: Please provide more detail, including sporozoite source and number, incubation times and temperatures, cell numbers, washes, etc. Was the infectious inocculum composed of ground, infected tick salivary glands or free sporozoites? Did you try the procedure using whole PBMC to generate an infected line, followed by magnetic bead sorting to pick out infected B cells, or did you only try to infect isolated B cells alone?
Figure 3: Include annotation of size
Line 302: Very confusing, re-word
Author Response
Revision report of reviewer 2:
Comments and Responses:
The entire questions are welcomed and answered accordingly. All major and minor changes are mentioned in red color text in the main text file. We will be available for further clarification.
Point 1:
My main reservation about this study is that the results were generated using a cloned line. In the closely related organism, T. parva, studies have shown that different T. parva infected, cloned lines produce different sets of cytokines. Thus, it is very likely that the results obtained by the authors using one B cell clone do not apply to T. annulata-infected B cells in general, especially in vivo, where infected cells interact with the rest of the immune system. Results would be strengthened if a large number of clones were studied, or if a several polyclonal B cell lines were analyzed.
Response: First, the study was performed in vitro where there was no interaction of cell line with other immune cells.
Second, the surface marker (CD21) on poly- and monoclonal cell line were present ~98% which indicates the same origin of cells (line no. 315-318; Fig. 6).
Thirdly, we already analyze and there was no significant difference in cytokine production between the mono- and polyclonal B cell lines of same passage, indicating that these derived from the same origin, which was confirmed by surface marker expression in these cell lines (Fig. 6). Thus, only monoclonal B cell line cytokines are mentioned in this manuscript.
The results of cytokine production might be different if we did cloning of transformed cells at 3rd passage (CD21 83%). But, we magnetically reisolated these 83% CD21+ cells (Fig. 5) at 4th passage to attain purity of ~99% (CD21+) which was successfully achieved (line no 300-301). These pure transformed cells were serially diluted to clone a cell for establishment of cell line following cytokines production (line no 171-172).
The detail is mentioned from line no. 369-372 of revised text file.
Point 2:
The rationale for the experiments using LPS is unclear, and the discussion of the results for these sections does little to put the information into perspective. Extensive revision is needed to clarify why these experiments were done, and what the results contribute to the scientific community in terms of immunology, parasitology, vaccinology, etc.
Response: Basically, LPS is being used as immune stimulator. In this study, LPS was used to compare its effect on nBC and TaBCs for cytokines production (line no. 194-195).
Secondly, LPS- and cytokine (IL-5 and IL-10)-stimulated B cells are associated with the production of natural immunoglobulins (IgM and IgG) in response to antigens. These associations served as the basis for using LPS to stimulate nBCs and TaBCs to compare their cytokines production abilities and further investigate immunological changes (immunoglobulin production) in these cell types. As LPS was not recognized and no immune response was observed by TaBCs. Hence, we direct future studies to observe the effects of various supplemental recombinant cytokines from other immune cells on these transformed cells (line no 473-475).
Point 3:
The grammar needs moderately extensive revision.
Response: Before submission the article was linguistically improved by American journal experts. But unfortunately the improvement was not significant. We claim to re-edit the whole text file that was well improved and an editing certificate is included at the end of text file. We hope that you can find it better now.
Point 4:
The abstract in the manuscript file does not match the one provided on the reviewer form.
Response: No option was found to replace during resubmission.
Point 5:
Line 51 is incorrect. Theileria parva infects both B and T lymphocytes.
Response: Is corrected in line no. 50-51.
Point 6:
Line 90 is incorrect. IL-10 is generally regarded as an anti-inflammatory cytokine, and IL-6 behaves in both pro-inflammatory and anti-inflammatory ways.
Response: Is corrected in line no. 90 and 91.
Point 7:
Line 100: How can evaluation of a single, clonal, line provide a comprehensive overview of cytokine production by T. annulata infected B-cells?
Response: There was no significant difference in cytokines production from polyclonal and monoclonal B cell line. Hence, only monoclonal cell line cytokines were mentioned in MS. In this way it provides a comprehensive overview of cytokine production by T. annulata infected B-cells.
The detail is mentioned from line no. 369-372 of revised text file.
Point 8:
Section 2.2: Please include information on veterinary care provided to infected animals and severity of clinical signs exhibited.
Response: Detail of veterinary care is included in line no. 133-136.
Point 9:
Section 2.4: Please provide more detail, including sporozoite source and number, incubation times and temperatures, cell numbers, washes, etc. Was the infectious inocculum composed of ground, infected tick salivary glands or free sporozoites? Did you try the procedure using whole PBMC to generate an infected line, followed by magnetic bead sorting to pick out infected B cells, or did you only try to infect isolated B cells alone?
Response:
B cells (CD21+) from experimentally T. annulata-infected cattle were magnetically isolated and culture them until their transformation and proliferation following surface markers identification at 3rd passage. Magnetically, reisolated CD21+ transformed cells, few cells were serially diluted for monoclonal cell line (line no. 169-171). Polyclonal and monoclonal cell line surface markers were again identified to confirm purity (line no. 175-177) following their cytokines production.
The detail of ticks to infect animals is included from line no. 131-132. Cell culture conditions and duration are mentioned from line no. 165-168.
The detail of sporozoite is mentioned in line no. 164.
Point 9:
Figure 3: Include annotation of size
Response: Annotations of DNA marker size were added in figure 3 and 7. To minimize the size of figure 8, results of 9 types of cytokines were adjusted in one slide.
Point 10:
Line 302: Very confusing, re-word
Response: Figure 4 legend was re-worded.

Round 2
Reviewer 2 Report
Regarding my previous Point 7, and the response (below):
Point 7:
Line 100: How can evaluation of a single, clonal, line provide a comprehensive overview of cytokine production by T. annulata infected B-cells?
Response: There was no significant difference in cytokines production from polyclonal and monoclonal B cell line. Hence, only monoclonal cell line cytokines were mentioned in MS. In this way it provides a comprehensive overview of cytokine production by T. annulata infected B-cells.
The detail is mentioned from line no. 369-372 of revised text file.
Thank you for adding this detail. However, I think the manuscript would be strengthened if you added a figure showing the lack of difference between the polyclonal and monoclonal B cell lines. The figure could be supplementary.
Author Response
Revision report of reviewer 2 round 2:
Comments and Responses:
Thank you for your valuable suggestion and comments for the improvement of manuscript. The demand for supplementary file is provided. The minor changes are mentioned in green color text in the main text file. We will be available for further clarification. Fourth affiliation of co-author Muhammad Adeel Hassan was missing, added in text file.
Point 7:
Line 100: How can evaluation of a single, clonal, line provide a comprehensive overview of cytokine production by T. annulata infected B-cells?
Response: There was no significant difference in cytokines production from polyclonal and monoclonal B cell line. Hence, only monoclonal cell line cytokines were mentioned in MS. In this way it provides a comprehensive overview of cytokine production by T. annulata infected B-cells.
The detail is mentioned from line no. 369-372 of revised text file.
Re-comment on Point 7:
Thank you for adding this detail. However, I think the manuscript would be strengthened if you added a figure showing the lack of difference between the polyclonal and monoclonal B cell lines. The figure could be supplementary.
Response: The figure is provided with the name of supplementary file.
The minor changes in lines no. 373-374, 393, 486, and 490-492 are mentioned in green color text.
